# WienerFlow: Wiener-Adaptive Flow Matching for Perception and Fidelity Trade-off in Low-light Image Enhancement

## Abstract

Low-light image enhancement (LLIE) strives to restore visibility and faithful details from severely under-exposed photographs. Existing learning-based approaches largely polarize around two objectives: fidelity-driven models, optimized for distortion metrics (e.g., PSNR, SSIM), tend to produce over-smoothed results with detail loss in extreme darkness, whereas perception-driven generative models synthesize visually appealing textures at the risk of hallucination. We bridge this dichotomy through **WienerFlow**, a continuous-time, flow-matching framework that unifies both objectives within a single linear transport path. Leveraging the theory of neural ordinary differential equations, we show that (i) a noise-free linear path originating from the low-light image equates to a fidelity-oriented trajectory, while (ii) a linear path initialized from Gaussian noise inherently favors perceptual richness. Under mild regularity assumptions, we prove that convex combinations of these two vector fields yield another valid linear flow, and we derive an optimal weight that maximizes perceptual realism subject to a fidelity budget. Extensive experiments on four LLIE benchmarks demonstrate that WienerFlow achieves state-of-the-art PSNR/SSIM scores while substantially improving perceptual quality, as confirmed by LPIPS, FID and NIQE on no-reference dataset, without introducing spurious textures. Our findings provide both a theoretical lens and a practical solution for balancing perception and distortion in low-light enhancement.

## 1 Introduction

Images captured in low-light environments or under extremely short exposure time often face challenges such as poor visibility, low contrast, color distortion and high noise levels. Low-light image enhancement (LLIE) aims to recover visually pleasing and information-rich images from severely underexposed inputs. Although convolutional neural networks (CNNs) and Transformer-based approaches have driven notable advances, the majority of learning-based LLIE methods are trained under fidelity-oriented objectives—optimizing Peak Signal-to-Noise Ratio (PSNR) or Structural Similarity Index (SSIM)—whose correlation with human visual preference is, at best, imperfect (Blau & Michaeli, 2018; Zhang et al., 2018). As a consequence, enhanced outputs often exhibit overly smooth transitions and substantial detail loss in extremely dark regions. Furthermore, constrained by the multi-step iterative nature of the inference process in diffusion models, these methods often require substantial inference time In contrast, generative paradigms, such as adversarial learning and particularly the recently prominent methods based on diffusion models (Jiang et al., 2021; Hou et al., 2024) emphasize perceptual realism and can synthesize plausible fine-scale structures in shadows. However, in the absence of explicit, physically or statistically grounded guidance, these models are prone to producing unrealistic textures or structural inconsistencies. This tension reflects the broader perception–distortion dilemma: improving perceptual quality typically compromises metric fidelity, and vice versa (Blau & Michaeli, 2018).

Recently, continuous-time generative modeling via neural ordinary differential equations (Neural ODEs) and flow matching has provided a rigorous framework for learning deterministic or stochastic transport maps between probability distributions (Chen et al., 2018; Lipman et al.,

2023). When LLIE is cast as transporting the low-light image distribution to its normal counterpart, the choice of the path's starting distribution fundamentally biases the learned transformation.

Specifically, (i) starting directly from the low-light image without injected noise induces a *fidelity-driven* path that preserves measured content yet lacks generative richness (Jung et al., 2025); (ii) initiating from Gaussian noise and conditioned on low-light images encourages a *perception-driven* path that can synthesize details but risks hallucination (Hou et al., 2024; Jiang et al., 2024).

To reconcile these competing desiderata, we introduce **WienerFlow**, a flow-matching paradigm that explicitly blends perception and fidelity oriented trajectories. We show that, under mild regularity assumptions, two linear flow paths that share a common endpoint (the normal image) are additively composable: any convex combination of their vector fields yields another valid linear path from a newly defined virtual start point to the common endpoint. Moreover, we theoretically establish the existence of an optimal convex weight that maximizes perceptual realism subject to a fidelity constraint (or vice versa). This additive property allows WienerFlow to learn a single continuous path that simultaneously honors fidelity and perceptual quality. Extensive experiments on multiple LLIE benchmarks demonstrate that WienerFlow attains competitive or superior PSNR/SSIM while markedly improving perceptual realism and reducing texture hallucinations compared with both direct mapping and purely generative methods. Our contributions are threefold:

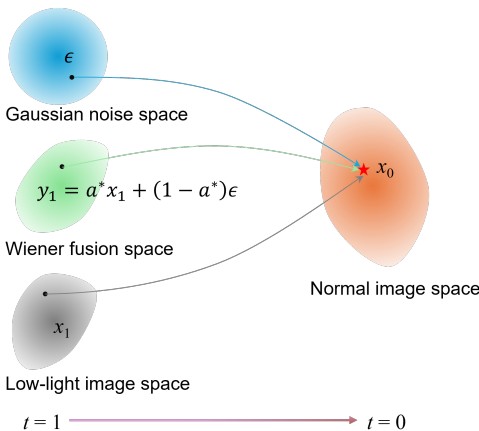

Figure 1: **WienerFlow: linear path bridges fidelity and perception.** The *fidelity* path (gray) directly transports the low-light distribution space to normal image space while the *perception* path (blue) transports Gaussian Noise to normal image space. $x_0$ sampled form the normal image distribution is the common endpoint of two flow-matching trajectories mentioned above. By linear additivity, a convex combination of their start points yields another valid linear path (green) starting at $y_1 = a^* x_1 + (1 - a^*) \epsilon$ and evolving as $y_t = t y_1 + (1 - t) x_0$. Choosing the optimal pixel-wise weight $a^*$ balances distortion and realism—biasing toward $x_1$ in high-SNR regions and toward $\epsilon$ in low-SNR shadows—while preserving the same endpoint $x_0$.

- We reinterpret LLIE through the lens of flow matching, revealing how noise injection implicitly governs the perception–fidelity bias of continuous transport paths.

- We prove an additive-composition theorem for linear flow paths and derive an optimal convex weighting strategy that balances perceptual and fidelity objectives.

- We develop *WienerFlow*, a practical algorithm based on MeanFlow that achieves state-of-the-art trade-offs between visual realism and distortion metrics across diverse datasets.

- We propose a time-length aware consistency loss, enabling even 1-step evaluation to achieve competitive performance.

## 2 RELATED WORKS

**Fidelity Driven Learning Based Methods.** Early deep learning approaches for LLIE largely superseded traditional methods by leveraging the power of large-scale data. Many of these initial works (Lore et al., 2017; Ren et al., 2019) employed direct end-to-end learning, utilizing Convolutional Neural Networks (CNNs) to learn a direct mapping from low-light to normal images. Concurrently, a significant body of works (Wei et al., 2018; Wu et al., 2022; Cai et al., 2023) drew inspiration from the Retinex theory (Rahman et al., 2004). Despite achieving significant progress over traditional techniques, these methods often rely on optimizing for fidelity-based loss functions (L1 or L2 loss). While models trained with these objectives excel at producing high Peak Signal-to-Noise Ratio (PSNR) and Structural Similarity Index Measure (SSIM) (Wang et al., 2004) scores, they often fall short in terms of human perceptual quality. The resulting images, particularly in extremely

dark scenes, tend to be overly smooth, sacrificing fine-grained details and textures that are crucial for visual realism.

**Generative Models for Perceptual Enhancement.** To address the shortcomings of fidelity-centric optimization, researchers turned to generative models, which are better suited for producing perceptually convincing results. Generative Adversarial Networks (GANs) (Goodfellow et al., 2014) were first applied to LLIE. EnlightenGAN (Jiang et al., 2021) stands out as a seminal work in this area. However, GAN-based methods are notoriously difficult to train, often suffering from instability, mode collapse, and the generation of undesirable artifacts, which has limited their widespread adoption and performance. Recently, Denoising Diffusion Probabilistic Models (DDPMs) (Ho et al., 2020) and Flow Matching Models (FM) (Lipman et al., 2022) have achieved state-of-the-art results across numerous image generation tasks. Their ability to produce diverse and high-quality images has motivated their application to LLIE. These models operate by learning to reverse a gradual noising process. By iteratively denoising a random noise map conditioned on the low-light input, they can generate a corresponding high-quality, normal image. Several recent works have demonstrated the impressive potential of diffusion models for LLIE. Methods such as GSAD (Hou et al., 2024) and LLDifffusion (Wang et al., 2025) have shown that diffusion-based approaches can restore stunning details and textures that were previously unattainable. Other works like ExposureDiffusion (Wang et al., 2023) aim to provide controllable enhancement by conditioning the diffusion process on an exposure value. Furthermore, some methods explore domain-specific adaptations, such as using a wavelet-based diffusion process (Jiang et al., 2023) to better capture frequency-domain information and saving computing cost.

However, these methodologies exhibit two primary drawbacks. First, while they effectively leverage the generative capabilities of diffusion models, the absence of explicit and principled guidance often leads to the generation of unrealistic textures. Consequently, the perceptual quality in darker regions is degraded, indicating a neglect of fidelity. Second, the inherent nature of diffusion models necessitates a multi-step sampling process during inference. Despite the application of acceleration strategies such as Denoising Diffusion Implicit Models (DDIM) (Song et al., 2020), a typical number of function evaluations (NFE) still exceeds ten steps, which curtails the practical applicability of these models.

## 3 METHODOLOGY

### 3.1 PRELIMINARY

**Flow Matching and MeanFlow Models.** Flow Matching (FM) offers a straightforward way to transform a simple prior $\epsilon \sim p_{\text{prior}}$ into the expected data distribution $x \sim p_{\text{data}}$ by prescribing a *velocity field* that drives latent particles along a continuous path $z_t = a_t x + b_t \epsilon$ in time $t$. The *marginal (instantaneous) velocity* is the conditional expectation over all microscopic flows:

$$v(z_t, t) = \mathbb{E}_{p_t(v_t|z_t)}[v_t], \qquad (1)$$

where $v_t = \dot{z}_t$ is the (sample-dependent) conditional velocity. A simple yet effective canonical path is the linear path: $a_t = 1 - t, b_t = t$. In this case, the velocity field can be expressed as $v_t = \epsilon - x$. Sampling is obtained by integrating the ordinary differential equation (ODE):

$$\frac{\mathrm{d}z_t}{\mathrm{d}t} = v(z_t, t), \qquad (2)$$

starting from $z_1 = \epsilon$ and running the flow backwards to $t = 0$. When using a Euler solver, the solution of each step can be obtained with $z_t = z_{t-1} + \Delta_t \cdot v(z_{t-1}, t)$, where $\Delta_t$ represents the discretized time interval.

**MeanFlow.** While CFM models the *instantaneous* field $v$, MeanFlow (Geng et al., 2025) introduces the *average* velocity over a finite interval $(r, t)$:

$$u(z_t, r, t) = \frac{1}{t - r} \int_r^t v(z_\tau, \tau) \, \mathrm{d}\tau. \qquad (3)$$

This quantity aligns with the net displacement $(t - r)u$ and depends jointly on the start and end times. Crucially, differentiating the definition yields the *MeanFlow identity*, an exact algebraic link

between average and instantaneous velocities,

$$u(z_t, r, t) = v(z_t, t) - (t - r) \frac{\mathrm{d}}{\mathrm{d}t} u(z_t, r, t), \tag{4}$$

which collapses to $u = v$ as $r \to t$. equation 4 is used to compute the ground truth average velocity during the traninig. In practical implementations, the first term in equation 4, which represents a constant velocity field, can be directly computed from the sampled image and noise. The second term, which involves taking the partial derivative of the current network with respect to time $t$, can be calculated using the Jacobian-vector product (JVP) operator in PyTorch.

## 3.2 WIENER-ADAPTIVE FUSION PATH

Let $x_0 \sim p_{normal}$ denote an image sampled from the well-exposed reference images, $x_1 \sim p_{low}$ is the low-light version of $x_0$, and $\epsilon \sim \mathcal{N}(0, I)$ is an i.i.d. Gaussian noise sample. All continuous paths are parameterised by $t \in [0, 1]$. Throughout, $\dot{x}_t \triangleq \frac{\mathrm{d}}{\mathrm{d}t} x_t$. We start from two linear (affine) paths that share the origin $x_0$:

$$\text{Fidelity path}: \quad x_t = t\, x_1 + (1 - t)\, x_0, \tag{5}$$

$$\text{Perception path}: \quad z_t = t\, \epsilon + (1 - t)\, x_0. \tag{6}$$

The flow path in equation 5 starts from the low-light observation, optimizing such a path approximated by a neural network $f(x_t, t; \theta_F)$ is fidelity oriented since it is anchored to the actual observation and enforces data-consistency. The path in equation 6 is perception oriented because starting from noise endows the model with generative capability, whose velocity estimator $f(z_t, t, x_1; \theta_P)$ is typically conditioned on low-light observation.

**Proposition 1** (Linear additivity). *For any real constants $a, b$, the mixture $y_t \triangleq a\, x_t + b\, z_t$ is itself an affine path in $t$ with closed-form*

$$y_t = t\,[a\, x_1 + b\, \epsilon] + (1 - t)\,(a + b)\, x_0. \tag{7}$$

*Consequently, $y_t$ satisfies the constant-velocity $\dot{y}_t = y_1 - y_0$, where $y_1 = ax_1 + b\epsilon$, $y_0 = (a + b)x_0$.*

*Proof.* Substituting equation 5–equation 6 and collecting terms:

$$y_t = a\,[tx_1 + (1 - t)x_0] + b\,[t\epsilon + (1 - t)x_0]$$
$$= t\,[ax_1 + b\epsilon] + (1 - t)\,(a + b)\, x_0,$$

which is affine in $t$; differentiation yields the claimed constant velocity. $\square$

**Corollary 1** (Endpoint preservation). *$y_t$ passes through the common start point $x_0$ iff*

$$a + b = 1 \quad \Longleftrightarrow \quad b = 1 - a. \tag{8}$$

*Under this necessary and sufficient condition the path reduces to*

$$y_t = t\,[a\, x_1 + (1 - a)\, \epsilon] + (1 - t)\, x_0, \tag{9}$$

*with new endpoint $y_1 = a\, x_1 + (1 - a)\, \epsilon$.*

**Wiener-adaptive fusion path.** We start the flow from a fusion endpoint that mixes the low-light observation and a noise draw:

$$y_1(i) = a(i)\, x_1(i) + [1 - a(i)]\, \epsilon(i), \qquad \epsilon(i) \sim \mathcal{N}(0, 1), \ \epsilon(i) \perp x_1(i), \tag{10}$$

with $x_1(i) = s(i) + n(i)$ and possibly heteroscedastic variances $\sigma_s^2(i) = \mathrm{Var}[s(i)]$ and $\sigma_n^2(i) = \mathrm{Var}[n(i)]$. To preserve generative diversity while avoiding fidelity bias, we align conditional means:

$$\mathcal{J}(a(i)) \triangleq \mathbb{E}\Big[\big(\mathbb{E}[y_1(i) \mid x_1(i)] - \mathbb{E}[s(i) \mid x_1(i)]\big)^2\Big]. \tag{11}$$

Intuitively, $(1 - a)\epsilon$ supplies diversity for perception, whereas equation 11 constrains the endpoint's conditional center to be fidelity-consistent in expectation.

**Proposition 2** (Expectation-aligned Wiener weight). *Minimizing equation 11 yields the pixel-wise optimal fusion weight*

$$a^*(i) \; = \; \frac{\sigma_s^2(i)}{\sigma_s^2(i) + \sigma_n^2(i)} \; = \; \frac{\text{SNR}(i)}{\text{SNR}(i) + 1}, \qquad \text{SNR}(i) \triangleq \frac{\sigma_s^2(i)}{\sigma_n^2(i)}. \tag{12}$$

*Proof of Proposition 2 can be found in the Appendix.* The result in 2 indicates that minimizing the conditional expectation leding to a Wiener fusion weight.

**MeanFlow for LLIE with Wiener-adaptive fusion path.** We use the additive linear path of equation 9 with a Wiener-adaptive fusion weight to obtain a *single* transport trajectory that balances perception and fidelity without training two separate models. Concretely, for each pixel $i$ we approximate the optimal fusion weight by

$$a^*(i) = \frac{\text{SNR}(i)}{\text{SNR}(i) + 1}, \quad \text{SNR} \approx \frac{G * \hat{s}^2}{G * \hat{n}^2 + \delta}, \tag{13}$$

where $G$ is a Gaussian smoothing operator, $\hat{s} = G * x_1$, $\hat{n} = x_1 - \hat{s}$ and $\delta$ is a small number to ensure numerical stability. The pixel-wise virtual starting point can then be formed with:

$$y_1 \; = \; a^* \odot x_1 \; + \; (1 - a^*) \odot \epsilon, \qquad \epsilon \sim \mathcal{N}(0, I), \tag{14}$$

and define the linear flow $y_t \; = \; t \, y_1 + (1 - t) \, x_0, t \in [0, 1]$, which shares the endpoint $x_0$ with the fidelity and perception paths but needs only one learned velocity field.

We adopt MeanFlow (Geng et al., 2025) as the flow-matching backbone to reduce function evaluations at inference. Let $u = f\big(y_t, r, t, x_0; \theta\big)$ denote the average velocity, estimated by an UNet parameterized by $\theta$ over a finite interval $(r, t)$, where $x_0$ is concatenated with $y_t$ along the channel dimension as a conditional guide. Times $t$ and $r$ are embedded by a shared time encoder and summed to yield the final time code. We train $u_\theta$ to align with the target average velocity obtained by the MeanFlow identity (equation 4), in our context the stantaneous velocity is $v = y_1 - x_0$ and the partial derivative of $u_\theta$ with respect $t$ is obtained via a JVP operator. Time sampling strategy of $(r, t)$ follows MeanFlow, sampled from LogitNormal$(-0.4, 1.0)$ with half of the minibatch enforcing $t = r$, this sampling scheme is denoted as $\mathcal{S}$. The training and inference algorithm can are shown in Alg. 1 and Alg. 2 respectively. The distance meteric in Alg 1, denoted as $d(\cdot, \cdot)$, is the Pseudo-Huber loss (Song & Dhariwal, 2023).

---

**Algorithm 1** Training

---

1: **while** not converged **do**
2:     sample a minibatch $\{(x_0, x_1)\}$ with $x_0 \sim p_{\text{normal}}, x_1 \sim p_{\text{low}}$
3:     sample $(r, t) \sim \mathcal{S}$ and $\epsilon \sim \mathcal{N}(0, I)$
4:     $a^* \leftarrow$ equation 13
5:     $y_1 = a^* \odot x_1 + (1 - a^*) \odot \epsilon$
6:     $y_t = t \, y_1 + (1 - t) \, x_0$
7:     $v = y_1 - x_0$
8:     $u = f(y_t, r, t, x_0; \theta)$
9:     $u_{\text{target}} = v - (t - r) \dfrac{\partial u}{\partial t}$
10:    $L = d\big(u, \text{stopgrad}(u_{\text{target}})\big)$
11:    perform a gradient descent step on $\nabla_\theta L$
12: **end while**

---

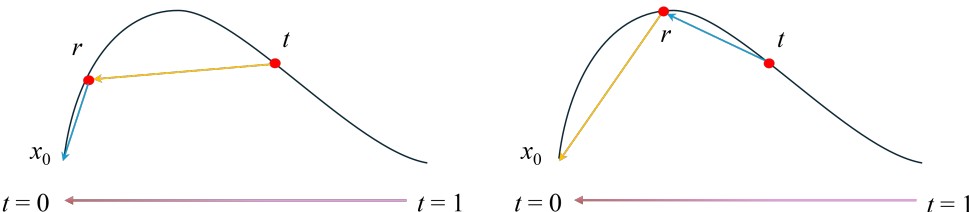

Figure 2: Tail-endpoint consistency. A teacher–student scheme stabilizes the shorter segment and learns the longer one; when $t \leq 2r$ the teacher (blue) is on $[t, r]$, otherwise on $[r, 0]$.

---

**Algorithm 2** Inference

1: **Inputs:** $x_0$ (reference/anchor), $x_1$ (low-light), trained $f(\cdot; \theta)$, steps $K$ (e.g., $K \in \{1, 2\}$)
2: Sample $\epsilon \sim \mathcal{N}(0, I)$
3: $a^* \leftarrow$ *equation* 13
4: $y_1 = a^* \odot x_1 + (1 - a^*) \odot \epsilon$
5: Initialize $z \leftarrow y_1, t = 1, r = 1 - \frac{1}{K}$
6: **for** $k = 1$ **to** $K$ **do**
7:    $u = f(z, r, t, x_0; \theta)$
8:    $z \leftarrow z - \frac{1}{K} u$
9:    $t \leftarrow t - \frac{1}{K}, r \leftarrow r - \frac{1}{K}$
10: **end for**
11: **Return:** $z$

---

### 3.3 TAIL-ENDPOINT CONSISTENCY FOR ONE-STEP QUALITY

**Motivation.** MeanFlow allows one-step generation from any time $\tau$ to 0 by evaluating $u(z_\tau, 0, \tau)$; in particular, $u(z, 0, 1)$ yields a direct one-step sampler. However, as the time interval enlarges, the approximation error of the mean field $u$ grows and degrades one-step fidelity and realism. Inspired by Consistency Models (Song et al., 2023), we design a length-aware consistency loss, which enforces that for any sampled time triplet $(0, r, t)$, the path integral starting from $t$, passing through $r$, and reaching the tail-endpoint $t = 0$ remains consistent and aligns with the ground truth $x_0$. Moreover, we analyze the relationship between the error in the velocity field of MeanFlow and the time length, with which to improve the consistency loss. To align with the notation adopted in MeanFlow, we denote the intermediate state by $z_t$ instead of $y_t$.

**Length-amplified mean-field error.** Recalling the MeanFlow identity (Eq. (4)):

$$u(z_t, r, t) = v(z_t, t) - (t - r) \, \partial_t u(z_t, r, t). \tag{15}$$

Let $u^*, v^*$ be ground-truth fields and $u_\theta, v_\theta$ their learned counterparts. With uniform gaps on the training support $E_u(r, t) \triangleq \sup_z \|u_\theta - u^*\|$, $E_v(t) \triangleq \sup_z \|v_\theta - v^*\|$, $E_{\partial u}(r, t) \triangleq \sup_z \|\partial_t u_\theta - \partial_t u^*\|$, subtracting the two versions of equation 15 and taking norms yields

$$E_u(r, t) \leq E_v(t) + (t - r) \, E_{\partial u}(r, t), \tag{16}$$

i.e., the mean-field error scales *linearly in the first order* with the interval length $(t - r)$. Consequently, the longer of $[t, r]$ and $[r, 0]$ is the *error-dominant* segment for single-step updates.

**Tail-endpoint consistency loss.** For any $0 < r < t \leq 1$, the velocity field over time satisfies with

$$z_t - x_0 = \int_0^t v(z_\tau, \tau) \, \mathrm{d}\tau = \int_0^r v(z_\tau, \tau) \, \mathrm{d}\tau + \int_r^t v(z_\tau, \tau) \, \mathrm{d}\tau. \tag{17}$$

Combining equation 17 with the average velocity defined in equation 3 leading to the triplet-based endpoint estimator,

$$\tilde{x}_0(t, r; \theta_1, \theta_2) \triangleq z_t - (t - r) \, u_{\theta_1}(z_t, r, t) - r \, u_{\theta_2}(z_r, 0, r), \tag{18}$$

where $\theta_1 = \theta_2 = \theta$, refers to the parameters of a model that performs well in multi-step iterations. Directly enforcing the estimation in equation 18 matches the real endpoint $x_0$ lead to a consistency

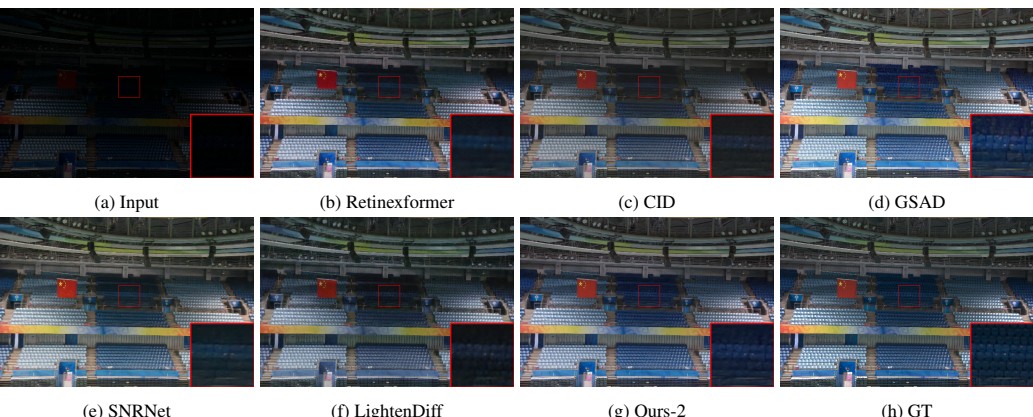

|  |  |  |  |
|---|---|---|---|
| (a) Input | (b) Retinexformer | (c) CID | (d) GSAD |
| (e) SNRNet | (f) LightenDiff | (g) Ours-2 | (h) GT |

Figure 3: Visual comparisons with other SOTA methods on real dataset LOLv2-real.

loss $\mathcal{L}_{(}t, r; \theta, \theta^{-}) = \mathbb{E}\left[d\big(\tilde{x}_0(t, r; \theta, \theta)\,,\,x_0\big)\right]$. To reflect the error dominance implied by equation 16, we assign the *longer* segment to the student parameters $\theta$ (receiving gradients) and the *shorter* segment to a frozen EMA teacher $\theta^{-}$ (no gradients). Let $\Delta_1 = r$ and $\Delta_2 = t - r$. The piecewise loss is

$$\mathcal{L}_{\text{TEC}}(t, r; \theta, \theta^{-}) = \begin{cases} \mathbb{E}\left[d\big(\tilde{x}_0(t, r; \theta^{-}, \theta)\,,\,x_0\big)\right], & \text{if } t \leq 2r \ \ (\Delta_1 \geq \Delta_2), \\ \mathbb{E}\left[d\big(\tilde{x}_0(t, r; \theta, \theta^{-})\,,\,x_0\big)\right], & \text{if } t > 2r \ \ (\Delta_2 > \Delta_1), \end{cases} \quad (19)$$

where $\mathbb{E}\left[\cdot\right]$ denotes the expectation over all random variables. Fig. 2 illustrates the composition of the consistency loss function under different time sampling distributions.

## 4 EXPERIMENTS

### 4.1 EXPERIMENTAL SETTINGS

**Datasets and Metrics.** We evaluate on three paired (full-reference) benchmarks and four unpaired (no-reference) collections. For full-reference evaluation we use LOLv1 (Wei et al., 2018), LOLv2-real, and LOLv2-synthetic (Yang et al., 2020). LOLv1 provides paired low/normal-light images captured under controlled exposure changes. LOLv2 extends this to more diverse scenes, with a real subset (LOLv2-real) of paired photographs and a synthetic subset (LOLv2-syn) generated from high-quality references by simulating low illumination. For no-reference evaluation we follow common LLIE practice and report results on DICM (Lee et al., 2013), LIME (Guo et al., 2017), MEF, and NPE (Wang et al., 2013), which contain natural low-light photographs without ground-truth references. We use the model trained on LOLv2-syn dataset and set NFE=2 to evaluate on no-reference datasets. On paired datasets we report fidelity metrics PSNR and SSIM (Wang et al., 2004) together with perceptual metrics LPIPS (Zhang et al., 2018) (lower is better) and FID (lower is better), computed between the enhanced outputs and the corresponding normal-light references. On the unpaired sets we use NIQE (Mittal et al., 2012) (lower is better) as a no-reference perceptual quality indicator.

**Baselines.** We compare against representative non-learning and learning methods across four families (see Table 1 and Table 2 for the complete list and citations): (i) *Traditional/Retinex:* MF (Fu et al., 2016a), LIME (Guo et al., 2017), SRIE (Fu et al., 2016b). (ii)*CNN/Transformers/Mamba:* Zero-DCE (Guo et al., 2020), RUAS (Liu et al., 2021), SCI (Ma et al., 2022), SNRNet (Xu et al., 2022), Retinexformer (Cai et al., 2023), MBTaylorV2 (Jin et al., 2025), CID (Yan et al., 2025). (iii) *Generative models:* EnlightenGAN (Jiang et al., 2021), LLFlow (Wang et al., 2022), PyDiff (Zhou et al., 2023), GSAD (Hou et al., 2024), LightenDiff (Jiang et al., 2024), LLDiffusion (Wang et al., 2025). We report two WienerFlow variants: Ours-1 uses a single sampling step (NFE= 1) and Ours-2 uses two steps (NFE= 2).

**Implementation details.** We implement all models in `PyTorch`. Training runs on a single NVIDIA RTX 4090 GPU. We use the Adam optimizer with its default $\beta$ parameters and a fixed learning rate of $1 \times 10^{-4}$. after 300k iterations we add the proposed tail-endpoint consistency loss

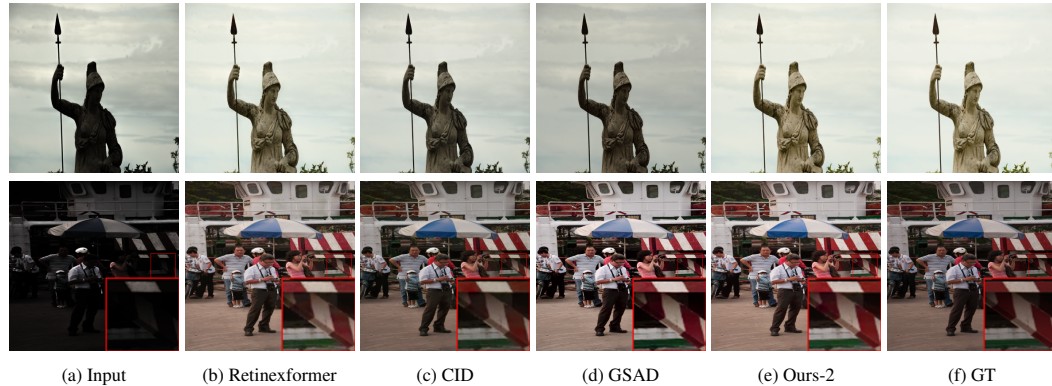

Figure 4: Visual comparisons with different methods on LOLv2-syn dataset.

Table 1: Quantitative results on LOLv1 (Wei et al., 2018), LOLv2-real (Yang et al., 2020), and LOLv2-synthetic (Yang et al., 2020) dataset. The best results are emphasized in **bold**, while the second best results are marked with an underlined.

| Methods | LOLv1 | | | | LOLv2-real | | | | LOLv2-syn | | | |
|---|---|---|---|---|---|---|---|---|---|---|---|---|
| | PSNR↑ | SSIM↑ | LPIPS↓ | FID↓ | PSNR↑ | SSIM↑ | LPIPS↓ | FID↓ | PSNR↑ | SSIM↑ | LPIPS↓ | FID↓ |
| MF (SP'16) | 16.97 | 0.508 | 0.380 | 119.23 | 18.73 | 0.627 | 0.388 | 96.22 | 17.49 | 0.792 | 0.185 | 52.54 |
| LIME (TIP'16) | 16.76 | 0.445 | 0.350 | 117.89 | 15.24 | 0.517 | 0.415 | 93.82 | 16.88 | 0.782 | 0.195 | 57.17 |
| SRIE (CVPR'16) | 11.86 | 0.495 | 0.257 | 107.68 | 14.45 | 0.612 | 0.312 | 87.86 | 14.50 | 0.672 | 0.241 | 73.26 |
| Zero-DCE (CVPR'20) | 14.86 | 0.562 | 0.335 | 101.23 | 18.06 | 0.680 | 0.312 | 91.95 | 17.76 | 0.838 | 0.168 | 49.24 |
| RUAS (CVPR'21) | 16.40 | 0.701 | 0.270 | 112.37 | 18.37 | 0.731 | 0.310 | 87.15 | 16.55 | 0.665 | 0.364 | 76.93 |
| SCI (CVPR'22) | 14.78 | 0.618 | 0.339 | 87.17 | 17.30 | 0.632 | 0.308 | 76.93 | 15.42 | 0.763 | 0.233 | 61.20 |
| SNRNet (CVPR'22) | 24.61 | 0.840 | 0.151 | 55.12 | 21.47 | 0.872 | 0.157 | 58.76 | 24.13 | 0.944 | 0.056 | 19.96 |
| Retinexformer (ICCV'23) | 25.15 | 0.843 | 0.131 | 71.15 | 22.79 | 0.866 | 0.171 | 62.46 | 25.67 | 0.952 | 0.059 | 22.75 |
| MBTaylorV2 (TPAMI'25) | 22.99 | 0.891 | 0.124 | 56.42 | 20.97 | 0.868 | 0.159 | 56.23 | 24.53 | 0.945 | 0.061 | 23.65 |
| CID (CVPR'25) | 23.49 | 0.870 | 0.105 | 52.51 | 23.42 | 0.862 | 0.169 | 50.17 | 25.70 | 0.942 | 0.047 | 19.00 |
| EnlightenGAN (TIP'21) | 17.48 | 0.652 | 0.275 | 98.49 | 18.63 | 0.730 | 0.309 | 92.57 | 16.57 | 0.802 | 0.212 | 74.32 |
| LLFlow (AAAI'22) | 21.14 | **0.904** | 0.119 | 64.58 | 17.43 | 0.846 | 0.176 | 77.05 | 23.42 | 0.950 | 0.050 | 20.79 |
| PyDiff (IJCAI'23) | **25.64** | 0.849 | 0.142 | 69.78 | 23.44 | 0.833 | 0.208 | 71.54 | 25.13 | 0.927 | 0.098 | 29.36 |
| GSAD (NeurIPS'23) | 22.34 | 0.897 | 0.110 | 57.63 | 20.11 | 0.865 | 0.113 | 47.49 | 24.13 | 0.942 | 0.052 | 19.36 |
| LightenDiff (ECCV'24) | 20.45 | 0.803 | 0.192 | 65.72 | 22.73 | 0.876 | 0.166 | 78.29 | 21.51 | 0.899 | 0.154 | 57.24 |
| LLDiffusion (PR'25) | 20.28 | 0.896 | **0.098** | **42.32** | 18.54 | 0.861 | 0.109 | 48.39 | 23.96 | 0.952 | 0.040 | 17.47 |
| Ours-1 | 20.47 | 0.854 | 0.136 | 68.82 | 23.24 | 0.882 | 0.108 | 40.46 | **26.05** | 0.958 | 0.042 | 15.61 |
| Ours-2 | 21.79 | 0.891 | 0.108 | 51.35 | 23.17 | **0.903** | **0.095** | **38.16** | 25.90 | **0.959** | **0.038** | **14.36** |

from Sec. 3.3 and continue training until convergence. Unless otherwise noted, reported results are obtained with the same pretrained model and the specified number of sampling steps (Ours-1 or Ours-2).

## 4.2 COMPARISON WITH OTHER SOTA METHODS

**Results on LOLv1 and LOLv2 datasets.** Table 1 summarizes results on the three paired benchmarks. Across datasets, Ours-2 delivers the strongest overall *perceptual* quality under fast sampling: on LOLv2-real it attains the best SSIM/LPIPS/FID (0.903/0.095/38.16); on LOLv2-syn it achieves the best SSIM/LPIPS/FID (0.959/0.038/14.36). On LOLv1, Ours-2 secures the second-best FID (51.35) and a top-3 LPIPS (0.108), trailing LLDiffusion (0.098) and CID (0.105). In terms of distortion, PSNR leadership largely resides with PyDiff on LOLv1/LOLv2-real (25.64/23.44,dB) and is competitive on LOLv2-syn, where our Ours-1 variant reaches the highest PSNR (26.05,dB) and Ours-2 is close behind (25.90,dB). Fig. 3 showcase the visual comparasions on LOLv2-real dataset, unlike other methods, our method suppresses artifacts and better preserves seat textures and edges, yielding natural contrast and colors. Visual results from LOLv2-syn dataset shown in Fig. 4 illustrate that our method tends to produce more natural color and brightness in dark aeras.

**One vs. two steps.** Moving from Ours-1 (NFE=1) to Ours-2 (NFE=2) consistently improves perceptual metrics with minimal change in PSNR: on LOLv1, LPIPS/FID improves $0.136 \rightarrow 0.108$ and $68.82 \rightarrow 51.35$; on LOLv2-real, LPIPS/FID improves $0.108 \rightarrow 0.095$ and $40.46 \rightarrow 38.16$ with SSIM

| (a) Ablation on fusion strategies. | | | |
|:---:|:---:|:---:|:---:|
| $a$ | $b$ | **PSNR** ↑ | **LPIPS** ↓ |
| 1 | 0 | 20.12 | 0.147 |
| 0 | 1 | 19.76 | 0.139 |
| $x_1$ | $1-x_1$ | 20.52 | 0.133 |
| $a^*$ | $1-a^*$ | **23.11** | **0.111** |

| (b) Ablation on the consistency loss $\mathcal{L}_{\text{TEC}}$ (NFE= 1). | | |
|:---:|:---:|:---:|
| **Setting** | **PSNR** ↑ | **LPIPS** ↓ |
| w/o $\mathcal{L}_{\text{TEC}}$ | 23.48 | 0.122 |
| $\mathcal{L}'_{\text{TEC}}$ | 23.29 | 0.121 |
| $\mathcal{L}_{\text{TEC}}$ (**ours**) | 23.24 | **0.108** |

Table 3: Ablations studies on LOLv2-real dataset.

$0.882 \rightarrow 0.903$; on LOLv2-syn, LPIPS/FID improves $0.042 \rightarrow 0.038$ and $15.61 \rightarrow 14.36$ while SSIM slightly increases $0.958 \rightarrow 0.959$ (PSNR $26.05 \rightarrow 25.90$dB).

**Results on no-reference datasets.** Table 2 reports NIQE on DICM, LIME, MEF, and NPE. WienerFlow achieves the lowest (best) NIQE on three of four datasets: LIME (3.33), MEF (3.46), and NPE (3.22). Relative to the second best results, this corresponds to 10.2% (LIME; $3.71 \rightarrow 3.33$), 6.7% (MEF; $3.71 \rightarrow 3.46$), and 9.3% (NPE; $3.55 \rightarrow 3.22$) reductions. On DICM, WienerFlow attains 3.30 NIQE and is competitive with the best published score (3.08), remaining within **7.1%**. These no-reference results corroborate the paired-set findings: Wiener-Flow improves natural image statistics in low-light scenes without introducing perceptual artifacts.

Table 2: Quantitative results on no-reference datasets in terms of NIQE (Mittal et al., 2012) on DICM (Lee et al., 2013), LIME (Guo et al., 2017), MEF, and NPE (Wang et al., 2013). The NIQE (Mittal et al., 2012) metric is assessed, with lower scores indicating better quality.

| Method | DICM | LIME | MEF | NPE |
|:---|:---:|:---:|:---:|:---:|
| LIME | 3.67 | 4.37 | 4.37 | 3.98 |
| Zero-DCE | 4.58 | 5.82 | 5.82 | 4.53 |
| EnlightenGAN | 4.06 | 4.59 | 4.59 | 3.99 |
| SNRNet | 6.12 | 3.76 | 3.76 | 6.44 |
| Retinexformer | **3.08** | 3.91 | 3.91 | 3.63 |
| GSAD | 3.28 | 4.32 | 4.32 | 3.55 |
| LightenDiffusion | 3.72 | 3.71 | 3.71 | 3.62 |
| Ours | 3.30 | **3.33** | **3.46** | **3.22** |

### 4.3 ABLATION STUDY

We conduct experiments on LOLv2-real to validate the effectiveness of our method. **Effectiveness of the proposed Wiener-adaptive fusion weight.** we compare several ways of forming the endpoint: (1) a path initialized purely from the low-light observation ($a=1$), (2) a path initialized purely from noise ($a=0$), and (3) a heuristic pixel-wise fusion that directly uses the low-light image as the weight (smoothed by a Gaussian for stability). Table 3a reports the results. The first two rows verify our intuition and theory: the *noise* path yields better perceptual quality (lower LPIPS) whereas the *observation* path provides better fidelity (higher PSNR). In contrast, our Wiener-adaptive weight $a^*$ simultaneously improves both distortion and perception, achieving the best PSNR/LPIPS among all variants.

**Effect of the consistency objective.** We further evaluate one-step inference variants to isolate the effect of the proposed temporal expectation-consistency loss $\mathcal{L}_{\text{TEC}}$. As shown in Table 3b, adding $\mathcal{L}_{\text{TEC}}$ consistently improves perceptual quality (LPIPS drops from $0.122$ to **0.108**) with negligible changes in PSNR, outperforming both training without the consistency term and a length-agnostic variant $\mathcal{L}'_{\text{TEC}}$. These results confirm that enforcing conditional-mean alignment along the short trajectory strengthens one-step perception while maintaining distortion.

## 5 CONCLUSION

We propose a Wiener-adaptive fusion endpoint and a single transport trajectory for low-light enhancement, initializing $y_1 = a^* x_1 + (1 - a^*)\epsilon$. An expectation-aligned objective yields the pixel-wise weight $a^* = \text{SNR}/(\text{SNR} + 1)$, preserving generative diversity while anchoring fidelity. One limatation of our work is thay SNR estimation may be biased under extreme noise/ISP mismatch; future work will explore learned calibration and content-adaptive trajectories.

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

## A PROOF AND ANALYSIS

**Centering (nonzero means).** Write $s(i) = \mu_s(i) + \tilde{s}(i)$ and $x_1(i) = \mu_s(i) + \tilde{s}(i) + n(i)$. Equivalently, one may include an intercept in the endpoint:

$$y_1(i) = \mu_s(i) + a(i)\big(x_1(i) - \mu_s(i)\big) + \big[1 - a(i)\big]\epsilon(i), \tag{20}$$

with $\epsilon(i) \perp (x_1(i), \tilde{s}(i), n(i))$ and $\mathbb{E}[\epsilon(i)] = 0$. All results below hold verbatim; for brevity we present the zero-mean notation and add $\mu_s(i)$ back as an intercept if desired.

**Lemma 1** (Conditional mean of the endpoint). *Let* $y_1(i) = a(i)\,x_1(i) + \big[1 - a(i)\big]\epsilon(i)$ *with* $\epsilon(i) \perp x_1(i)$ *and* $\mathbb{E}[\epsilon(i)] = 0$. *Then*

$$\mathbb{E}\big[y_1(i) \mid x_1(i)\big] = a(i)\,x_1(i). \tag{21}$$

*Proof.*

$$\mathbb{E}\big[y_1(i) \mid x_1(i)\big] = \mathbb{E}\big[a(i)\,x_1(i) + \big(1 - a(i)\big)\epsilon(i) \mid x_1(i)\big] \tag{22}$$
$$= a(i)\,x_1(i) + \big(1 - a(i)\big)\,\mathbb{E}\big[\epsilon(i) \mid x_1(i)\big] \tag{23}$$
$$= a(i)\,x_1(i), \tag{24}$$

where independence implies $\mathbb{E}[\epsilon(i) \mid x_1(i)] = \mathbb{E}[\epsilon(i)] = 0$. $\qquad\square$

*Proof of Proposition 2.* By Lemma 1, we have

$$\mathbb{E}\big[y_1(i) \mid x_1(i)\big] = a(i)\,x_1(i). \tag{25}$$

Let $X \triangleq x_1(i)$ and $m(X) \triangleq \mathbb{E}\big[s(i) \mid X\big]$. The objective in equation 11 becomes

$$\mathcal{J}\big(a(i)\big) = \mathbb{E}\Big[\big(a(i)\,X - m(X)\big)^2\Big], \tag{26}$$

i.e., the $L^2$ distance between $m(X)$ and the linear function $a\,X$ under the marginal of $X$. The orthogonal projection of $m(X)$ onto $\mathrm{span}\{X\}$ yields the slope

$$a^*(i) = \frac{\mathbb{E}\big[m(X)\,X\big]}{\mathbb{E}\big[X^2\big]}. \tag{27}$$

Using the tower property and $X = s(i) + n(i)$ with $\mathbb{E}\big[s(i)\,n(i)\big] = 0$, we get

$$\mathbb{E}\big[m(X)\,X\big] = \mathbb{E}\Big[\,\mathbb{E}\big[s(i) \mid X\big]\,X\,\Big] \tag{28}$$
$$= \mathbb{E}\big[s(i)\,X\big] \tag{29}$$
$$= \mathbb{E}\big[s(i) \cdot \big(s(i) + n(i)\big)\big] \tag{30}$$
$$= \mathbb{E}\big[s^2(i)\big] + \mathbb{E}\big[s(i)n(i)\big] \tag{31}$$
$$= \sigma_s^2(i) + \mathbb{E}^2\big[s(i)\big] \tag{32}$$
$$= \sigma_s^2(i), \tag{33}$$
$$\mathbb{E}\big[X^2\big] = \mathrm{Var}(X) + \mathbb{E}^2\big[X\big] \tag{34}$$
$$= \mathrm{Var}(X) + \big(\mathbb{E}\big[s(i) + n(i)\big]\big)^2 \tag{35}$$
$$= \mathrm{Var}\big(s(i) + n(i)\big) \tag{36}$$
$$= \sigma_s^2(i) + \sigma_n^2(i). \tag{37}$$

Therefore,

$$a^*(i) = \frac{\sigma_s^2(i)}{\sigma_s^2(i) + \sigma_n^2(i)} = \frac{\mathrm{SNR}(i)}{\mathrm{SNR}(i) + 1}, \qquad \mathrm{SNR}(i) \triangleq \frac{\sigma_s^2(i)}{\sigma_n^2(i)}. \tag{38}$$

$$\square$$

**Gaussian specialization.** If $s(i) \sim \mathcal{N}(0, \sigma_s^2(i))$ and $n(i) \sim \mathcal{N}(0, \sigma_n^2(i))$ independently, then $\mathbb{E}[s(i) \mid x_1(i)] = \frac{\sigma_s^2(i)}{\sigma_s^2(i) + \sigma_n^2(i)}\,x_1(i)$, and the coefficient in Proposition 2 matches the Wiener/LMMSE shrinkage.

**Degenerate cases.** If $\sigma_s^2(i) = 0$ (locally constant signal), then $a^*(i) = 0$; if $\sigma_n^2(i) = 0$ (noiseless), then $a^*(i) = 1$. Both align with the intuition of pure prior vs. pure fidelity.

## B    NETWORK ARCHITECTURE

Table 4: Baseline UNet architecture used in our experiments.

| Component | Configuration |
|---|---|
| Input channels | 6 (concatenated low-light and noisy image) |
| Output channels | 3 (RGB enhanced image) |
| Inner channels | 64 |
| Number of downsampling stages | 4 |
| Middle block | 2 Residual blocks (1 with attention, 1 without) |
| Number of upsampling resolution | 4 |
| Channel multipliers per resolution | (1, 1, 2, 2, 4) |
| Self-attention resolution | 32 (128*128 as input) |
| Residual blocks per resolution | 2 |
| Normalization | GroupNorm (32 groups) |
| Time embedding | Positional encoding + 2-layer MLP |
| Downsampling | Strided Conv2d (kernel=3, stride=2, padding=1) |
| Upsampling | Nearest neighbor + Conv2d (kernel=3, padding=1) |
| Residual block type | Two Conv2d layers |
| Skip connection | Concatenate |
| Final convolution | Conv2d (kernel=3, padding=1) |

## C    COMPLEXITY ANALYSIS

We conduct complexity analysis on our method and compared with other diffusion based method. With only 1–2 NFEs, our models operate at a fraction of the compute of diffusion baselines (e.g., *Ours-2*: 84.12G FLOPs, 22.32M params), yet deliver superior perceptual quality on LOLv2-real (best SSIM/LPIPS/FID) and competitive or higher PSNR. *Ours-1* minimizes complexity (NFE=1) while retaining strong distortion performance, and *Ours-2* adds a modest cost to achieve state-of-the-art perception, yielding the most favorable accuracy–efficiency trade-off among compared methods.

Table 5: Efficiency and accuracy on **LOLv2-real**. Best is **bold**, second-best is underlined. NFE = # function evaluations. PSNR/SSIM/LPIPS/FID are from Table 1. FLOPs for **Ours-2** are 84.120G with 22.320M params.

|  | LLDiffusion | GSAD | LightenDiffusion | Ours-1 | Ours-2 |
|---|---|---|---|---|---|
| **FLOPs (G)** ↓ | 551.975 | 67.020 | 118.960 | **84.120** | 84.120 |
| **Params (M)** ↓ | 208.711 | **17.173** | 20.743 | 22.320 | 22.320 |
| **NFE** ↓ | 10 | 10 | 20 | **1** | 2 |
| **PSNR** ↑ | 18.54 | 20.11 | 22.73 | **23.24** | 23.17 |
| **SSIM** ↑ | 0.861 | 0.865 | 0.876 | 0.882 | **0.903** |
| **LPIPS** ↓ | 0.109 | 0.113 | 0.166 | 0.108 | **0.095** |
| **FID** ↓ | 48.39 | 47.49 | 78.29 | 40.46 | **38.16** |

## D    VISUAL RESULTS.

We showcase more visual results in this section.

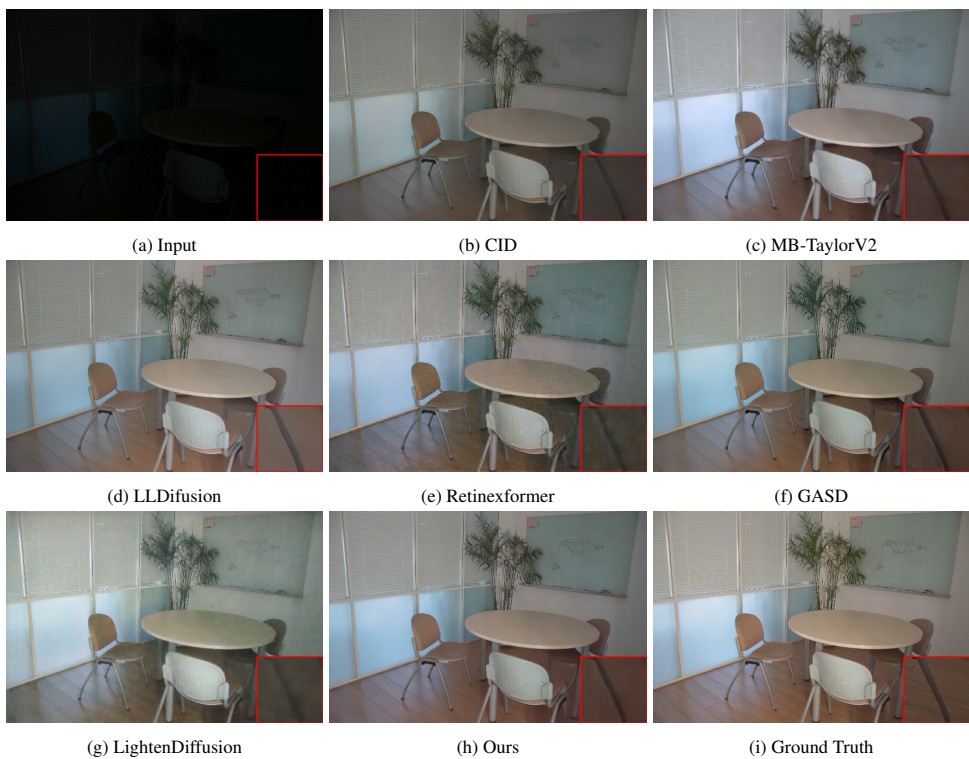

Figure 5: Visual comparisons on LOL-v2-real. The local patch has been zoomed out for improved visibility. Under extremely low-light conditions, the proposed method enhances brightness while maintaining realistic and natural color reproduction, outperforming other approaches.

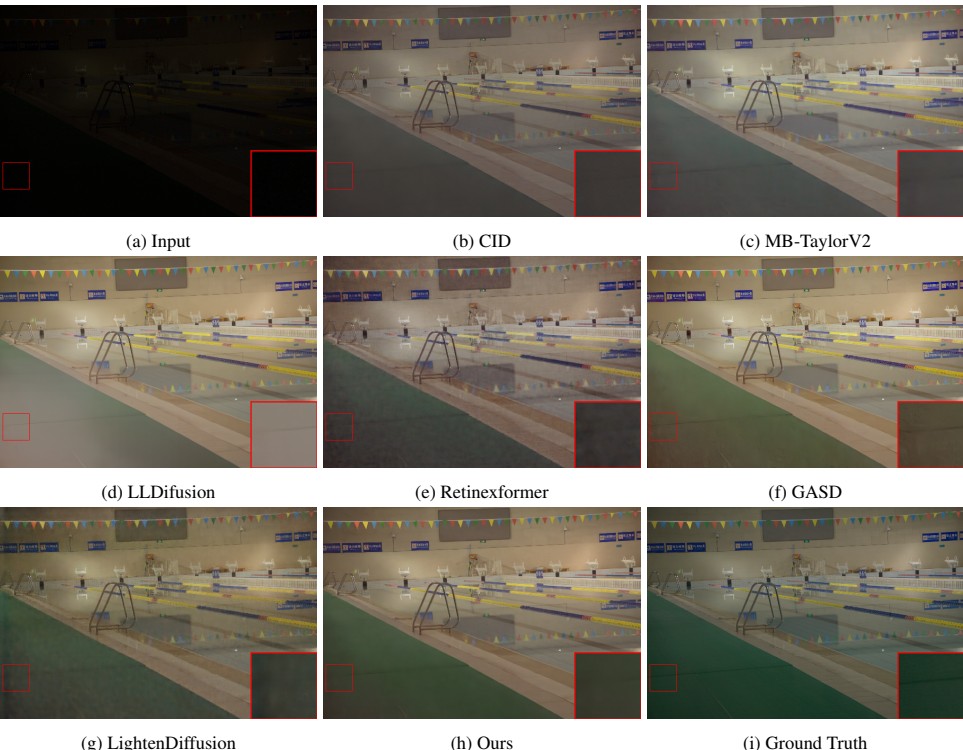

Figure 6: Visual comparisons on LOL-v2-real.

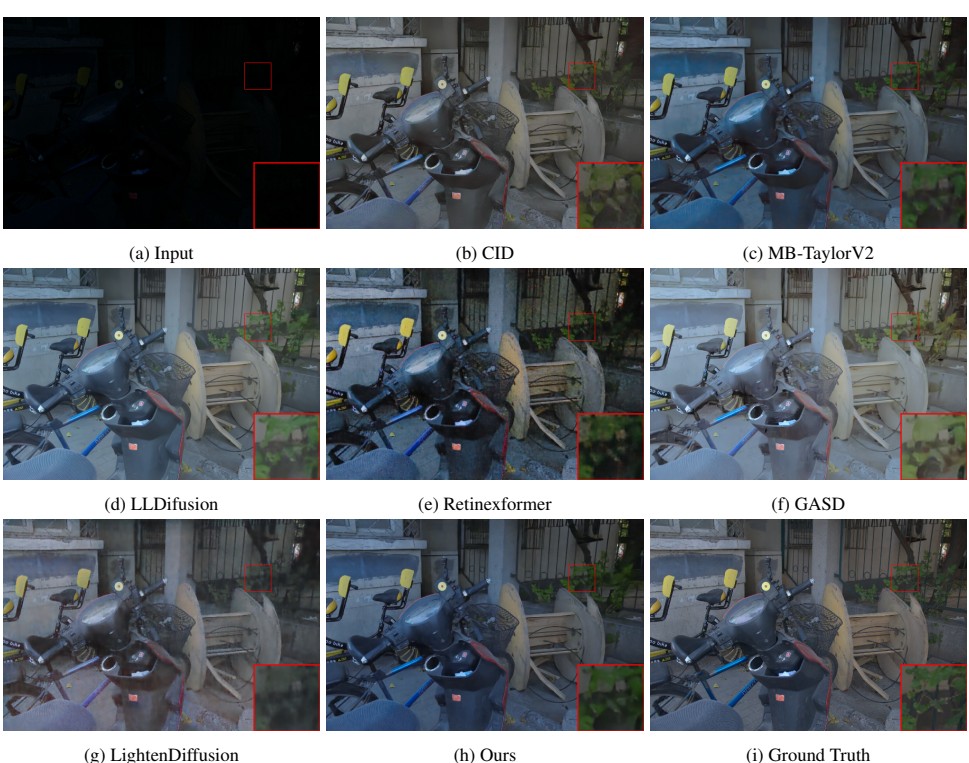

Figure 7: Visual comparisons on LOL-v2-real.

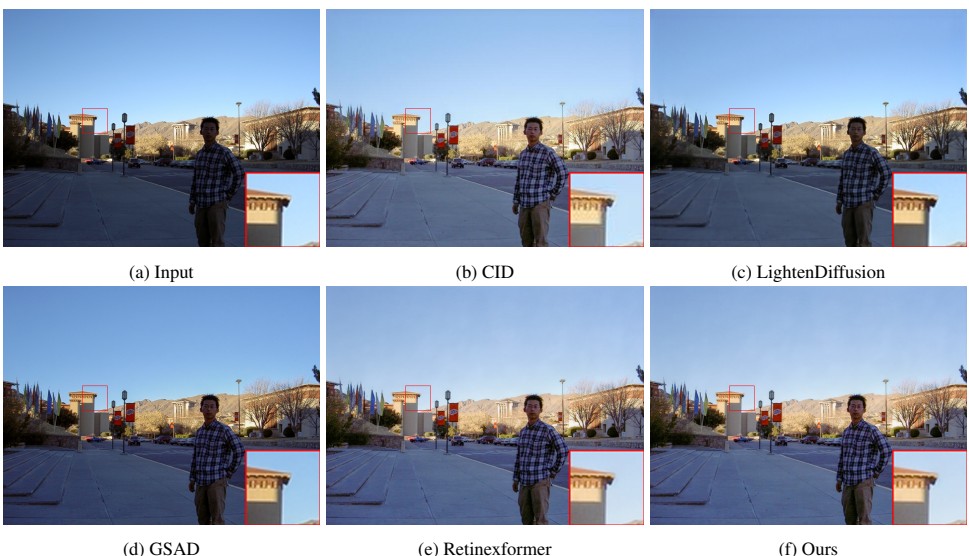

Figure 8: Visual comparisons on DICM.

