# OpenReview forum: "WienerFlow: Wiener-Adaptive Flow Matching for Perception and Fidelity Trade-off in Low-light Image Enhancement"
_ICLR.cc/2026/Conference — ICLR 2026 Conference Withdrawn Submission_

### Official Review · Reviewer_dvZJ · 2025-10-31

**Soundness:** 3
**Presentation:** 2
**Contribution:** 2
**Rating:** 4
**Confidence:** 3

**Summary:**

This paper addresses the long-standing perception-distortion trade-off in low-light image enhancement: fidelity-driven methods (optimized for PSNR/SSIM) produce over-smoothed results with detail loss in extreme darkness, while perception-driven generative models (e.g., diffusion, GANs) synthesize visually appealing textures but risk hallucinations. To bridge this gap, the authors propose WienerFlow—a continuous-time flow-matching framework based on neural ordinary differential equations (Neural ODEs). WienerFlow unifies two conflicting trajectories into a single linear transport path. Extensive experiments on 7 LLIE benchmarks (e.g., LOLv1/v2, DICM, LIME) show WienerFlow achieves state-of-the-art performance: it outperforms SOTA methods in perceptual metrics (LPIPS, FID, NIQE) while maintaining competitive PSNR/SSIM, with only 1–2 NFEs (≈84G FLOPs) for fast inference.

**Strengths:**

- The paper directly addresses the most practical pain point in LLIE—balancing "objective fidelity" (critical for industrial scenarios like surveillance, medical imaging) and "subjective perception" (essential for consumer applications like photography). Unlike prior work that prioritizes one objective over the other, WienerFlow provides a principled way to unify both.
- WienerFlow achieves a rare balance of performance and speed (1–2 NFEs, 22.32M parameters, 84G FLOPs).
- The experiments cover both paired and unpaired benchmarks, with comparisons across 4 method categories. Ablation studies validate the necessity of the Wiener-adaptive weight (Table 3a) and TEC loss (Table 3b), ensuring the framework’s components are not redundant.

**Weaknesses:**

- The paper adopts MeanFlow (a pre-existing flow-matching backbone, 2025) and applies it to the LLIE task by adding a Wiener-adaptive fusion path. From the current presentation, this design risks being viewed as a straightforward "A+B" combination rather than a deep innovation.
- The framework is exclusively validated on the LLIE task, with no exploration of its applicability to other image restoration scenarios—despite the fact that its core "perception-fidelity balance via flow-matching" logic could potentially be generalized. Key untested tasks include image super-resolution, inpainting, denoising.
- The paper uses MeanFlow as the flow-matching backbone but provides no ablation to justify this choice over other flow-matching variants (e.g., Conditional Flow Matching (CFM), Rectified Flow). It is unknown whether MeanFlow’s "average velocity" modeling (vs. CFM’s instantaneous velocity) is essential for WienerFlow’s performance.

**Questions:**

- Your work uses MeanFlow as the backbone—how does the Wiener-adaptive fusion path go beyond a simple "add-on" to modify or extend MeanFlow’s core logic for LLIE? For example, does the fusion path require adjusting MeanFlow’s velocity field calculation, or is there a unique interaction between the Wiener weight and average velocity that is specific to low-light enhancement?
- Have you conducted preliminary experiments on other image restoration tasks (e.g., inpainting, super-resolution)? If so, what modifications (if any) are needed for WienerFlow to adapt to these tasks? If not, do you believe the framework’s "perception-fidelity balance via flow-matching" logic is inherently limited to LLIE, or could it be generalized with minimal adjustments?

---

### Official Review · Reviewer_GR2j · 2025-10-31

**Soundness:** 3
**Presentation:** 2
**Contribution:** 2
**Rating:** 4
**Confidence:** 4

**Summary:**

This paper introduces a continuous-time, flow-matching framework for LLIE. Specifically, based on the observations that a flow from the low-light image to the clean image denotes a fidelity-oriented trajectory, and a flow from pure Gaussian noise to the clean image is a perceptual-favored path, this paper derives an optimal weight that maximizes perceptual realism subject to a fidelity budget. Experiments are conducted on several benchmark datasets.

**Strengths:**

1. The derivations and propositions in Equations 7–19 present a solid theoretical analysis for the proposed method.

2. This paper is well organized and presented with high quality.

**Weaknesses:**

- It seems only $y_1$ is used in the actual flow: $y_t = ty_1 + (1 - t)x_0$, $t \in [0, 1]$. If this is true, the Proposition 1 is actually not necessary, as only its endpoint is used and analyzed in the proposed method, thus leading to understanding difficulty for readers but providing no meaningful theoretical contributions.

- In $x_1(i) = s(i) + n(i)$, I assume $i$ denotes the pixel, but it's not clear what $s$ and $n$ mean. Signal or noise intensity? Also, why can Equation 13 be employed to calculate SNR? I will check the derivation here later, and I strongly suggest the authors to make a clearer discussion for this part.

- In Algorithm 2, $x_0$ is not available in the inference stage, making the inference algorithm quite weird.

- GPP-LLIE (AAAI'25), a diffusion-based LLIE method, should be discussed in related works and be compared in Table 1.

- Results of some baselines (Retinexformer) are borrowed from their original papers, while some results are different from those in the original papers. The authors are suggested to explain this inconsistency.

- I also hope the authors can clarify the difference between the proposed method and PMRF (ICLR 2025), especially the unconditional rectified flow model defined in the controlled experiments.

- Some claims in the abstract don't align with the experiments. For example, experiments are conducted on three benchmarks, however, the abstract claims 4 datasets. Besides, on no-reference datasets, only reporting NIQE is not sufficient to demonstrate WienerFlow's perceptual quality.
- According to the ablation, I don't see any trade-off between perception and fidelity.

**Questions:**

The original rating (4) is given in my original review because I hope the authors can provide a rebuttal for my questions. However, please note this is a generous rating. The quality of this paper is obviously higher than score 2, but many clarifications and revisions are still needed to meet my current score of 4.

For questions, please see the weakness part.

---

### Official Review · Reviewer_XDmz · 2025-11-02

**Soundness:** 2
**Presentation:** 2
**Contribution:** 2
**Rating:** 4
**Confidence:** 3

**Summary:**

The paper studies low‑light image enhancement (LLIE) via a flow‑matching formulation that explicitly interpolates between two endpoints: a fidelity‑driven path initialized from the observed low‑light image and a perception‑driven path initialized from noise. A single linear transport path is formed at a fused endpoint, and MeanFlow is trained to follow this path. The fusion weight is derived from a Wiener criterion, estimated per‑pixel using smoothed signal/noise statistics, so the model adaptively balances distortion and perceptual quality. To stabilize very low‑step sampling, the authors introduce a tail‑endpoint consistency loss (LTEC) that encourages agreement between short‑trajectory predictions and a teacher trajectory. Experiments on paired (LOLv1/2) and unpaired benchmarks (NIQE on several datasets) indicate competitive or improved perceptual metrics, with competitive PSNR/SSIM on standard LLIE sets.

**Strengths:**

- [S1] The idea is conceptually simple which is essentially a single linear flow that blends fidelity and perception.
- [S2] The paper draws a per‑pixel Wiener‑style weight to the method to SNR, offering a clear method for the perception–distortion trade‑off.
- [S3] Competitive quality at 1–2 NFE steps.

**Weaknesses:**

- [W1] Limited technical novelty. Proposition 1 is mathematically straightforward, and the Wiener weight is a classical result; the main contribution is packaging these into a MeanFlow‑based LLIE system plus an adaptation of consistency training. The paper slightly over‑emphasizes the theoretical component relative to the simplicity of the underlying identities.
- [W2] Ambiguous SOTA claims on distortion. The abstract states SOTA PSNR/SSIM broadly, but the main table shows the result of PSNR is mixed (e.g., PyDiff has the best result on LOLv1 and LOLv2‑real). The paper should calibrate claims to the tables.
- [W3] Ablations are under‑scoped. Fusion ablations are only on LOLv2‑real; there is no study of SNR approximation design (smoothing kernel, δ), noise proportion schedules, or sensitivity to the time‑sampling distribution used by MeanFlow, which are all central to the approach’s stability and quality.
- [W4] Runtime/NFE vs. wall‑clock comparisons to recent diffusion LLIE baselines are absent. The paper reports 1–2 steps but not throughput or latency on common hardware.

**Questions:**

1) For LTEC, could you add teacher‑decay sweeps (EMA rate) and an ablation replacing EMA with a fixed snapshot teacher? Any signs of instability without the piecewise assignment in Eq. (19)?
2) Please include a failure case discussion and a stress test (e.g., extreme low‑light/ISP mismatch), since SNR bias is noted as a limitation.

---

### Official Review · Reviewer_KBmn · 2025-11-02

**Soundness:** 2
**Presentation:** 2
**Contribution:** 2
**Rating:** 4
**Confidence:** 4

**Summary:**

This paper tries to improve the dark portions of images. The goal is to make them brighter and clearer, the trade-off being: making the image accurate to the original scene versus making it look naturally pleasing to our eyes. This paper finds a middle ground using WienerFlow.

**Strengths:**

The paper has a well-structured narrative. It starts with a clear explanation of the perception-distortion problem, introduces the core idea of fusing two paths, provides the mathematical foundation, and then validates it with extensive experiments.
The paper uses figures effectively for visualisation. Figure 1 is particularly good, providing an intuitive visual explanation of the complex "Wiener-adaptive fusion path" concept. The result figures (e.g., Fig. 3, 4, 5) are clear and directly support the quantitative claims by showing superior detail and colour reproduction.
The "Wiener fusion" is the main concept. Using a pixel-wise SNR to balance between a "fidelity path" (starting from the dark image) and a "perception path" (starting from noise) is a mathematically grounded and elegant solution to a long-standing problem.
The mathematical derivations for the linear additivity (Proposition 1) and the optimal Wiener weight (Proposition 2) are presented clearly. They provide a solid theoretical foundation for the method.
The paper performs a very comprehensive evaluation. It compares against a wide range of methods (over 15 baselines) spanning different families: traditional methods, CNNs/Transformers, and other generative models (GANs, Diffusion). This makes the results highly convincing.
The use of multiple standard datasets (LOLv1, LOLv2-real, LOLv2-synthetic) for paired evaluation and several others (DICM, LIME, etc.) for unpaired evaluation is a best practice and allows for a complete assessment of performance.
The method achieves state-of-the-art results, especially in perceptual metrics (SSIM, LPIPS, FID), while maintaining competitive fidelity (PSNR).
Significant Efficiency Gain is a major contribution. By building on MeanFlow, WienerFlow requires only 1-2 function evaluations (NFE) compared to 10-20 for other diffusion-based models. This is a major improvement in inference speed, making high-quality generative LLIE much more practical, as shown convincingly in Table 5.

**Weaknesses:**

The inference algorithm uses the ground truth well exposed image as an input. The paper does not clarify how this is handled in a real-world scenario. This is a critical point for practical application that needs explanation.
While the paper includes an ablation study for the fusion weight in Table 3, it does not do the same for other key components.
More user studies for perception can strengthen the claim.
The entire method hinges on a good estimate of the local SNR. The paper does not discuss how sensitive the model is to errors in this estimation. A brief analysis or discussion on the robustness of this step would be a valuable addition.

**Questions:**

The inference algorithm uses the ground truth well exposed image as an input. The paper does not clarify how this is handled in a real-world scenario. This is a critical point for practical application that needs explanation.
While the paper includes an ablation study for the fusion weight in Table 3, it does not do the same for other key components.
More user studies for perception can strengthen the claim.
The entire method hinges on a good estimate of the local SNR. The paper does not discuss how sensitive the model is to errors in this estimation. A brief analysis or discussion on the robustness of this step would be a valuable addition.

---

### Note · Authors · 2025-12-03

I have read and agree with the venue's withdrawal policy on behalf of myself and my co-authors.